# Household Cooking and Eating out: Food Practices and Perceptions of Salt/Sodium Consumption in Costa Rica

**DOI:** 10.3390/ijerph18031208

**Published:** 2021-01-29

**Authors:** Adriana Blanco-Metzler, Hilda Núñez-Rivas, Jaritza Vega-Solano, María A. Montero-Campos, Karla Benavides-Aguilar, Nazareth Cubillo-Rodríguez

**Affiliations:** 1Costa Rican Institute of Research and Teaching in Nutrition and Health (INCIENSA), Tres Ríos 4-2250, Costa Rica; hnunez@inciensa.sa.cr (H.N.-R.); jvega@inciensa.sa.cr (J.V.-S.); mariamonteroc54@gmail.com (M.A.M.-C.); kbenavides@inciensa.sa.cr (K.B.-A.); 2Project Consultant, International Development Research Centre (IDRC), Llorente de Tibas 11303, Costa Rica; raquel.cubillo@gmail.com

**Keywords:** salt, salt reduction, sodium, practices, perceptions, household, eating out of home

## Abstract

This research aims to study the food practices and perceptions related to excessive consumption of salt/sodium when cooking and eating outside the home in a study population representing the wide intergenerational and sociocultural diversity of Costa Rica. Key communities from around the country, cultural experts, and key informants were selected. Four qualitative research techniques were applied. Data was systematized based on the Social Ecological Model. Women are generally in charge of cooking and family food purchases. Salt is perceived as a basic ingredient, used in small amounts that can be reduced—but not eliminated—when cooking. Changes in food preparations and emotions associated with the consumption of homemade food with salt were identified. The population likes to eat out, where the establishments selected depend mainly on age group and income. Beyond cultural and geographical differences, age aspects are suggested as being the main differentiators, in terms of use of salt, seasonings, and condiments in the preparation of food at home, the recipes prepared, and the selection of establishments in which to eat out. The deeply rooted values and meanings associated with salt in food indicate that the implementation of salt reduction strategies in Costa Rica is challenging.

## 1. Introduction

Food consumption is a social activity [1]. Far from being only biological or behavioral, this multifactorial phenomenon involves political, psychological, social, cultural, economic, and environmental aspects of high complexity. Beyond the need for nourishment, food consumption is loaded with meanings and emotions linked to social events that have nothing to do with the strict need to eat [2].

The association between excessive salt and sodium consumption with high blood pressure, the leading cause of death worldwide [3] and the most important modifiable risk factor for cardiovascular and cerebrovascular diseases, is one of the main implications of food in public health [4,5,6,7]. In 2017, high sodium intake led to approximately three million deaths and a loss of 70 million disability adjusted life-years [8].

In Latin America, the prevalence of hypertension in adults (>35 years old) varies between 26% and 42% [9]. In 2014, in Costa Rica, it was 36% (31% diagnosed and 5% nondiagnosed) in adults (>20 years old) and, since 1970, cardiovascular diseases have been the leading cause of death in the country [10].

Clinical trials and population studies have shown that reducing salt intake is associated with a reduction in the rate of hypertension and that this strategy is profitable. However, only a few countries have implemented successful interventions [11].

Sodium consumption and time trends have been estimated in the Costa Rican population (based on food and beverage acquisitions). Sodium available for consumption in national households largely exceeded the intake levels recommended by the World Health Organization (<2 g sodium/person/day) and increased between 2005 and 2013 from 3.9 to 4.6 g/per adult person/day (*p* < 0.001) [12]. The estimated mean sodium intake in Costa Rican children and adolescents from 7 to 18 years of age, based on a food frequency survey and added salt to the served food, was 3.4 g with a sodium density of 1.8 g of sodium/1000 kcal, maintaining significant differences by age subgroups (*p* < 0.001) [13].

The source of dietary salt varies according to the population group, the food system, the cooking and preparation methods of food in the household and out, and water sources for human consumption [14,15]. In developed countries, sodium comes mainly from processed foods [16]. Unlike Japan and China, where it generally comes from food cooked with soy and/or fish sauces [17], in Costa Rica and Brazil, sodium comes mainly from salt for domestic consumption (≥60%) and processed foods and condiments with added sodium (27.4%). Meanwhile, in younger populations, meats and fried chicken, alone or with “casado” (a typical dish) with soy or Lizano^®^ sauce (a commercial vegetable sauce similar to Worcestershire); pastries and sandwiches; fast food such as “caldosas” (a local snack preparation made with fish and fried corn chips); hamburgers; hot dogs; French fries; Ketchup or pink sauce (mix of Ketchup and mayonnaise) contributed to most of their sodium intake. These examples indicate that studies on food preparation and the use of salt should be contextualized, as they are modulated by historical, geographical, generational, social, and cultural elements [18].

A qualitative exploratory study to identify the knowledge, perceptions, and behaviors (KABs) related to the consumption of salt and sodium in food and its relationship to health was conducted during 2011 in urban and rural areas of Costa Rica, as well as two other Latin American countries. Most interviewees felt that food could not be consumed without salt. They did not know that processed food contains salt and sodium. Although they did not measure the amount of added salt in foods and understood that excessive consumption poses a health risk, the participants believed that they consumed little salt and did not perceive that their health was at risk. It was also found that there is a public awareness about salt, but not of “sodium”, and it was supposed that more sodium is consumed than what is reported, with no prospects of reducing its consumption [19]. Another study carried out at the same time described individual KABs regarding salt intake and its dietary sources in one city per country of five countries of the Americas, including Costa Rica [19]. Almost 90% of participants associated the excess intake of salt with the occurrence of adverse health conditions, more than 60% indicated they were trying to reduce their current intake of salt, and more than 30% believed reducing dietary salt to be of high importance. Only 47% of respondents stated they knew the salt content in food items [20].

This study is part of the National Plan to Reduce the Consumption of Salt/Sodium in Costa Rica 2011–2021 [21]. It is based on the need to supplement disciplinary scientific knowledge, international and institutional policies and initiatives related to the reduction of salt/sodium consumption, cerebrovascular mortality statistics (“the etic”), and what arises from local knowledge and the daily life of the population of the communities of Costa Rica (“the emic”). It is based on a qualitative pilot study carried out during 2011 by the same research team, in two communities within one region of the country, in order to confirm whether the findings on KABs related to the consumption of salt/sodium and health [19] were transferable to populations of other geographic and sociocultural regions of Costa Rica. It is considered a pioneer study, intended to identify the eating patterns and practices related to salt and sodium in diverse communities located throughout Costa Rica.

In this research, we aimed to study food practices and perceptions related to excessive consumption of salt and sodium, at home and outside the home, in communities from all regions of Costa Rica, including various cultural practices of the population’s diet. Findings derived from participant responses served as input to identify the sociodemographic profile of the target segment and design formative research questions, in order to build a social marketing and communication plan for reducing the excessive consumption of salt and sodium.

## 2. Materials and Methods

### 2.1. Study Design

The present observational study is anthropological–ethnological. Ethnography was used as a research method, as ethnology is a branch of anthropology. This method allowed us to observe the behaviors of the participants. It also allowed for recording their cultural practices, decisions, and actions related to food consumption and the intake of salt/sodium in their diets. Explicit efforts were made to understand food consumption practices from the individual participant’s perspective. Ethnographic investigations have great potential for the interpretive and reconstructive analysis of reality [22].

### 2.2. Research Techniques and Data Collection

In order to meet the goal of constructing a reliable and comprehensive picture, four research techniques were used: (1) a semistructured interview [23], (2) an investigative workshop [24], (3) a demonstration [23], and (4) participant observation [25]. They were applied as follows:

(1) Semistructured interview: the interviewee had the opportunity to express their points of view in an open remark interview situation. (2) Investigative workshop: offered the ability to explore the research topic from comprehensive and participatory perspectives through discussion, reflection, and group feedback. The purpose of these workshops was to complement and deepen the information obtained from the interviews with cultural experts and key informants from the communities. The workshops were conducted by an anthropologist, further facilitating the study investigation. Each workshop consisted of an introductory section, such as the explanation of dynamics and individual presentation, or the recapitulation of the previous workshop; a development section and a closing activity. (3) Demonstrations: were carried out by the participants who agreed to prepare dishes that they usually eat at home as a family. This allowed the researchers to verify what was reported during the interviews and workshops, and to identify other details surrounding culinary practices and family eating habits. (4) Participant observation: allowed the researchers to methodically systematize the information they observed, overheard, and otherwise gathered regarding daily practices and processes of salt/sodium consumption. Research team observations were made and recorded according to guides developed specifically for each technique, as well as the characteristics of the participants.

First, interviews were carried out and systematized with cultural experts. Workshops and interviews with key informants were then developed, and observations were made during both. Last, demonstrations were conducted, with those participants who agreed to perform them. The four techniques allowed a progression from the description of the data, to reflection and action, on through the prioritization of facts and needs, culminating in the formation of decisions supported by the majority of the participants.

The techniques are complementary to one another, enabling the collection of abundant, richly significant data, that when triangulated, validates the research results as pointed out by Maxwell (2002) [26].

### 2.3. Location of the Study

The study was carried out in 23 communities in Costa Rica, selected based on two criteria:(a)Location according to geographical region: the selection of locations for data collection was based on the regionalization proposed by the Ministry of National Planning and Economic Policy (MIDEPLAN), considering the dimensions of the economy, social participation, health, and education. The country consists of seven provinces divided into six regions: Central, Brunca, Chorotega, Huetar—Atlantic, Huetar—North, and Central Pacific. Most of the population (62%) resides in the Central Region, which was thus subdivided into four subregions: Central North (Heredia), Central East (Alajuela), Central West (San José), and Central South (Cartago) [21].(b)Statistics on cardiovascular mortality were provided by the Directorate of Health Surveillance of the Ministry of Health of Costa Rica [27]. Key communities were selected based on those with the highest prevalence of stroke mortality due to close association with salt intake [28]. See the list of communities and description of the geographical and sociocultural characteristics of the regions of MIDEPLAN in Appendix A.

### 2.4. Participants

Key informants were recruited by our cultural experts primarily via, by the following criteria: ages between 20 and 65 years; sex (female and male); minimum residency time in the community of 10 years; cultural roots (knowledge of the region’s food supply), and responsibility for home meal preparation, purchasing of food ingredients or doing both. In addition, snowball sampling increased the number of participants.

The cultural experts (managers) were social scientists with knowledge of the region and its culture. They were hired by the Ministry of Culture and Youth to work with the local counterparts of these communities, promoting processes associated with sociocultural management (e.g., heritage and artistic promotion, among others). Their work is aimed at co-ordination, production, participatory training, and research, in order to instill self-management capacities that allow each community to direct its own cultural development, well-being, and holistic health.

Due to the small population of our communities, the sample size for this qualitative study was determined by relying on past investigative projects in which optimal results were achieved by interviewing two individuals, and having group discussions of 6–8 participants (19). This saturation-based criterion was applied to each of the communities involved in our study.

The 91 participating adults comprised 11 cultural managers and 80 key informants (26 participated in interviews and 54 in 12 workshops).

In the present study, special attention was paid to the ingredients added during food preparation to increase taste and aroma. Salt, seasonings, and condiments were classified according to the Costa Rican context and the research objective. Common or discretionary salt was tested independently of other flavorings, such as seasonings and condiments. Seasonings were conceptualized as herbs and natural seasonings without added salt, such as onion, garlic, bell pepper, and cilantro, among others. Condiments were defined as those processed products with added salt, such as dehydrated bouillon powder, bouillon cubes, sauces, and dressings, among others.

### 2.5. Interviews and Workshops

The list of the questions (by study category) used in the participant interviews and their corresponding community workshops can be found in Appendix B.

The field research was conducted between February and July of 2016. It was carried out by an anthropologist with the assistance of social, nutritional, and food science professionals. The statements were recorded with a voice recorder and photographic images were taken in order to have evidence of the cultural material [23] related to behaviors, such as daily practices of acquisition, preparation, and consumption of food within sociocultural contexts, either inside or outside the home.

### 2.6. Data Analysis

The themes and concepts of the data were derived from the predefined categories in the previous exploratory study (19) and considered those emerging from the present study. For the systematization of qualitative data, we considered the theoretical–conceptual components, based on the Social Ecological Model [29,30]. All the collected information—90 documents between the interviews and research workshops—was transcribed by one individual and was then analyzed by the team anthropologist, using the ATLAS-ti 8 software (Scientific Software Development GmBH, Berlin, Germany). The research team periodically reviewed the qualitative information, which was enriched with the contributions of each of the researchers, until reaching a consensus on the interpretation of the results.

The analyzed information was classified into two types—household cooking and eating out—given that the food sources of sodium, and strategies to control excessive consumption differ for each of these scenarios. In addition, for food prepared at home, details on practices and perceptions to enhance flavor with (a) common salt and other salts and (b) seasonings and condiments were given, as they are the primary sources of sodium in Costa Rica [18]. Speech [31] and content analyses were applied [32].

### 2.7. Ethical Considerations

The Scientific Ethical Committee of the Costa Rican Institute of Research and Teaching on Nutrition and Health (INCIENSA) approved this observational study; the code is IC-2015-01. All participants voluntarily agreed to sign an informed consent form, including authorization to record conversation and capture photographs. At the time of the study, the Costa Rican National Health Research Council (CONIS) certified the researchers as observational and interventional researchers.

## 3. Results

### 3.1. Population Characteristics

The cultural managers were 55% men and 45% women with university degrees and professional training in the areas of social sciences, communication, or dramatic arts. They were over 30 years old and had worked for the office for more than 4 years (some exceeding 15 years). Through their work, cultural managers understand the sociocultural characteristics of the population, as they carry out activities focused on cultural management, artistic cultural expression, entrepreneurship, and the revitalization of (tangible and intangible) cultural heritage.

Of the community members, most (>61%) were middle-age adults (35–65 years) and women (88%) with different occupations. Most women were engaged exclusively with domestic work (>42%), followed by merchants and professionals (17% each) and students (13%). The remaining were operators and technicians.

The participants interviewed were additionally categorized. They had very different educative levels (32% with incomplete basic studies, 24% with completed high school studies, and 44% with university studies). Most reported belonging to lower and middle socioeconomic strata (33% and 67%, respectively), which was confirmed through the index created. Most (71%) reported their ethnic group as mixed ancestry (“mestizos”), the remaining being white (17%), indigenous (8%), or Afro-Caribbean (4%). Most reported not having high blood pressure (72%) and none as having noncommunicable diseases (NCD).

### 3.2. Food and Cooking in the Different Regions and Cultural Groups of Costa Rica

According to the statements of the participants and the observations made by the research team, the basic meal for lunch (noon) in the study communities was “casado”, which consists of rice and beans (black or red) with some protein of animal origin (meat, sausages, egg, and cheese), ripe banana, a salad consisting of cabbage and tomato, “picadillo” (chopped vegetables), and corn tortillas; the latter common mainly in the Chorotega Region. In all regions, the consumption of “gallo pinto” (a mixture of rice and beans with onion and Lizano^®^) was mentioned for breakfast. In the Chorotega Region, dinner was served with “gallo pinto” and corn tortillas with smoked cheese or “Bagaces”-type artisanal products with high sodium content.

The typical meals traditionally consumed on Sundays, holidays, and other special occasions were now less frequently made, or their preparation was adapted with the use of processed ingredients to reduce the time in the kitchen while still achieving the desired flavor. For example, in the families of old mothers in the Huetar Caribe Region, on Sunday, “rice and beans” were prepared with coconut milk, Panamanian chili and thyme, accompanied by fish or chicken in Caribbean sauce, cabbage salad, and fried plantain.

In general, food was prepared similarly in all communities (fried, stir-fried, braised, or stewed). Flavoring was carried out by adding common table salt, “olores” (onion, sweet chili, and garlic), and herbs (oregano, thyme), which had been partially displaced by bouillon cubes, dehydrated bouillon powder known as “consomé”, dry herb mixes with added salt, and commercial sauces to achieve the desired flavor, mainly in the younger generations. In all regions, table salt was added to food with the fingers, or sometimes a spoon. Participants indicated that people of indigenous origins (or their descendants) salted food for its preservation, a practice that is being lost.

The consumption of tropical fruits with salt, especially among the youngest participants, was present in all of the communities of the country, while Lizano^®^ sauce had become popular in recent years. The intake of fast food, which is rich in salt, has spread throughout the country, due to the demands of young people and tourism. International cuisine has been mainly introduced through tourism in coastal areas. Youth from the Central Region reported new trends in cooking (alternative, organic, vegetarian, vegan) or lifestyle (healthy).

### 3.3. Preparing and Consuming Food at Home

#### 3.3.1. The Acquisition and Preparation of Food

According to the population consulted, the acquisition of family food or grocery shopping was carried out conjointly by both parents (37%); just a father or mother (13% each); or sons and daughters, but in low proportion (12%); while the remaining 38% were a combination of father with daughter, mother with daughter, and all members of the family. The decision-making process was usually carried out jointly, with its variants according to the composition of the family nucleus.

Based on these observations, three types of buyer profiles were identified within the residents of the different participating regions: young people who had begun living independently, families that have minors among their members, and families where there are no minors.

As most participants were women of economically active age who work part-time or full time, mothers alone (54%) or with their daughters (13%) prepared and served meals during the night and morning before going to work. Just 12% of men reported cooking food at home (4% alone and 8% with the help of a daughter).


*“Do you cook every day at home? Who cooks?*



*Yes, we cook every day. Women are the ones who cook, and, on special occasions, men feel probably ashamed, sorry, men get involved. Hahaha”*
(Woman, Bagaces)

Participating mothers agreed that they have changed their taste for adding salt or condiments when preparing food. Furthermore, they declared that in the past, seasonings were used to cook, while commercial products are now added. The use of seasonings in combination with condiments is frequent during the preparation of food.


*“Look, here it is, this is the rice with chicken that we are going to prepare, we are going to add all kinds of condiments, vegetables..., we are going to add bell pepper, celery, cilantro and condiments, and a little salt. And then, here are the mayonnaises; the dressings that also have a little salt for taste and a few condiments, also to add a little flavor, we can add mustard. And here are the condiments that we are adding. It can be a few bouillon cubes, that’s what the lady says, we can add a little bit of Lizano^®^ sauce, mustard, we can add… what else can we add, it can be complete with Lizano^®^ sauce.”*
(Woman, Alajuela)

Participants pointed out that, while preparing food, they try the food and add more salt or condiments until they feel that it has a good flavor:


*“Because I add condiment, bouillon… Lizano^®^ sauce, all that—haha—I try it for salt with the tip of the spoon. If it’s salty I do not add any more, but I do if it needs a little. As I was saying, I only calculate it myself, a half teaspoon, a pinch, a teaspoon.”*
(Woman, Cartago)

Most participants believed they cook with only a little salt; one participant, for instance, indicated:


*“You taste it, you never know (…), add a little first and, then taste it. If you feel like you almost don’t taste salt, you add a little more, and so on, you continue tasting it.”*
(Woman, Liberia)

Most informants agreed that salt in excess is “bad” and exemplified it with phrases such as: “If you eat a lot of salt you become a donkey,” “It is salty,” and “Some salt fell on it.” Most of the interviewees showed no reluctance to reduce salt/sodium consumption during food preparation. However, there were reservations about family members who would complain when a meal was prepared with less salt and end up adding more salt to the dish served, as well as Lizano^®^ sauce, ketchup, mayonnaise, and/or dressings.

Data collected through interviews and observations confirmed that locals usually consumed instant soft drinks, breakfast cereals, cream sandwich cookies, sweet bread, canned sweet corn, canned fruit cocktails, and added ketchup in meals. In addition, depending on the taste, the population tended to balance sweet and sour flavors by adding salt to enhance the flavor. They consumed pineapple (*Ananas comosus*) and orange (*Citrus sinensis*), in natural state or juice, with salt. In addition, they reported eating mango with salt and lemon and, as observed in the parks of the communities, they sometimes add Lizano^®^ sauce. Additionally, they mentioned that sweet-tasting food and products do not contain salt/sodium.

When participants had celebrations at home, they said they commonly prepare other dishes with processed food and use a greater variety of condiments than those used daily. They also stated that they usually make “arroz con siempre” (a colloquial name referring to rice with chicken, typically served in celebrations and festivities in Costa Rica) with a side salad (usually beet salad or cabbage with chopped tomato), mashed beans, and chips; these are served with soda crackers on “special” occasions.

##### The Use of Discretional Salt and Other Salts

Most community participants stated that salt is a basic ingredient used to give flavor to foods. It is also used to give flavor to fruits such as orange, cas (*Psidium friedrichsthalianum*), guava (*Psidium guajava*), mango (*Mangifera indica*), and pineapple; to fruit drinks (orange juice); to alcoholic beverages, such as tequila (distilled liquor consumed with lemon and salt) and “michelada” beer (mixture of beer, lemon, and salt). Young adults pointed out that they would not stop consuming salt and will continue with the practices of directly adding salt or Lizano^®^ sauce to fruits.

Participants indicated that salt’s flavor is a taste they have acquired over time and that the amount of salt used has increased. Only a few of them reported adding salt to their food, but indicated that if their relatives perceive that the food does not have the desired amount of salt, they take it directly from the container (“jar or pot”) located in the kitchen and add it to the food served (only on their plate). They reported that the use of saltshakers on the table is rare.

Those responsible for preparing food in families reported not usually using standardized measures to add salt to the dishes but, instead, would use a spoon (for ease) or a pinch (amount of salt taken between the thumb and index finger) to get food to the desired level of saltiness.

Most of the mothers interviewed mentioned that they had become used to the SalSol^®^ salt brand in their homes. To improve the taste of foods, they also use other types of salt, such as sea salt, flavored or seasoned salts (e.g., garlic, onion, herb, lemon, and smoked). Some bought sea salt, as they believed it to be thicker and healthier, as it does not contain bleaching additives, while others tried low-sodium salt on the recommendation of health personnel, but did not like the taste and, so, stopped using it:


*“It was Naty who, when we were preparing hash for an event, told me to please get her the sea salt, and since then, I only buy that. Then I bought sea salt and this one. It is more expensive, but for me, it salts more with less quantity.”*
(Woman, San José)


*“Well, I try to buy this sea salt from a friend who goes to Lepanto, Puntarenas to bring salt, she brings this one and another one a little thicker, but it’s not as thin as common salt, it is not a specific brand. And if I don’t have this one, I buy SalSol.”*



*“I always look for... this one... I don’t know! I think this is the best one, coarser, well, let’s say it like that, it has like bigger grains. I believe the more refined the salt, the more harmful, you know, how they process it and all that. So, I think this is the best one... I always go to the supermarket and look for the one with bigger grains.”*



*“I generally use the coarse salt for meats, either broiled or baked. It is usually very used for that.”*
(Man, Heredia)

There is a community recognized in the country for the commercialization of pork rinds (fried pork meat), to which they add a different type of salt—curing salts (composed of sodium nitrites and nitrates)—which provide intense color and flavor to meats, such as pork meat:


*“Then, it depends on how it is done. My husband learned how to cook pork rinds from my sister’s husband, who in turned learned the recipe from Puriscal residents, and his pork rinds are delicious without being too salty. Another fact is that people from Puriscal and Mora use pink salt, they use it on the pork rinds to give them that color and taste that you cannot find elsewhere. My husband has even had one glass of that salt, bought from the butcheries in Puriscal. It is like salt contraband because it is what they use and it is one of Puriscal pork rind’s secrets. You can go eat pork rinds anywhere else and they will never taste like pork rinds from there.”*
(Woman, San Jose)

When key informants from the community were queried about salt fortification, it was evident they did not have any knowledge about it. Few of them knew that salt serves as a source of fluoride and iodine, although they recognized the contributions that iodine and fluoride make to the body separately.

##### The Use of Seasonings and Condiments

It was observed that, while preparing food, women mixed seasonings (e.g., onions, peppers, garlic, cilantro, celery, oregano, and rosemary, among others) with condiments, instant soup, dehydrated creams, and commercial sauces (Lizano^®^ sauce, mayonnaise, and tomato sauce). In a few cases, they just used a small amount of seasonings.

Most participating mothers indicated the use of condiments, such as bouillon powders (e.g., chicken, beef, or shrimp broth), dry-herb mixtures with added salt, commercial sauces (e.g., vegetable or Lizano^®^ sauce, tomato sauce, and soy sauce), and dressings (e.g., mayonnaise). A smaller group declared using bouillon cubes and flavored salts to give more flavor or to improve the taste of food during its preparation and to quickly achieve the flavor preferred by most of their family members, who judged whether the food needs more salt or flavor; or, on the other hand, consider that it was prepared with “love”.

Younger women (below 30 years) referenced new dish preparation practices, mainly due to the offer of products sold to create specific flavors. They added shrimp bouillon to rice to prepare rice with shrimp flavor. They used dehydrated soups with mushrooms, noodles, meat, or vegetables to prepare sauces for meats or pasta, and also used breading batter to prepare fish, seafood, and chicken.

Participating younger and older women, as well as men, indicated that the incorporation of new practices into food preparation related to the diversity of products in the market and the desire to give food the taste of international dishes. There were mentions of using soy sauce and oyster sauce to give food an Asian flavor.


*“Condiments, all of them, or seasonings better said and a little oyster sauce. And I do not add salt because it (oyster sauce) is salted already.”*
(Woman, Palmar Norte)


*“(…) mostly when they eat fried chicken, they put it on the fries and the one I use is oyster sauce.”*
(Woman, Palmar Norte)

Soy sauce, called Chinese sauce in Costa Rica is used in the preparation of fried rice (e.g., the rice served in Chinese restaurants):


*“I cook lots of fried rice. I boil it first, because rice needs to be boiled in order to be fried. Boiled rice without salt in it. I do not use salt when I am frying it; I put Chinese sauce in it. Chinese sauce is salty; therefore, no salt is required. Well, before I would put a little “ajinomoto” (local common name used for monosodium glutamate) in it. I no longer buy it for anything, but the Chinese sauce will salt it.”*
(Woman, Limon)

Another sauce that mainly young men mentioned as a red meat and chicken wing seasoning was BBQ sauce, either for grilling or frying. Its use was attributed to various goals:


*“(…) it gives food a very original taste, delicious.”*
(Investigative workshop, Guanacaste, man)


*“Those BBQ ribs; take the ribs and the wings and wash them well and put them to… (pauses) pour oil in a pan and put them in the oven. When they are almost done, before taking them out, pour the BBQ sauce. Then the BBQ sauce will give them flavor and you do not need to put anything else on them.”*
(Woman, Liberia)

Moreover, it was common to use a commercial vegetable sauce to prepare many dishes. One of the informants expressed:


*“For everything! Lizano^®^ sauce is used in all dishes here, it is almost indispensable, and I know it must have lots of sodium because it is pure condiment, right? It is used in “pinto”, in eggs…well not in white rice and beans, but certainly when recipes are prepared: “picadillo” (hash) or like the soup I am currently preparing, I will put a little Lizano^®^ sauce in it shortly. But yes, generally almost everything has Lizano^®^ sauce in it; well…it is English sauce…”*
(Man, Alajuela)

There was a diversity of commercial bouillon powder brands in the national market. However, people expressed a preference for one specific brand, which is well-positioned in the market:


*“Well, I do not know, I always buy Maggi^®^ because that is what my mom used to buy. I have never tried a different one, I always buy that one, let’s say bouillon powders…”*
(Woman, Alajuela)

As part of food preparation practices, some participants combined bouillon powders and domestic salt; others added bouillon powder instead of natural seasonings:


*“Many people do not know that if they work with spices, food is really good without putting bouillon powders on them; that is what salt is for. Nevertheless, people do not realize it. People think that if they do not use bouillon powder, their food will not taste good.”*
(Woman, Guanacaste)

The participants of the workshop expressed that the consumption of Maggi^®^ products had prevailed from generation to generation, for its use in chicken and beef rib broths, as well as instant soups and dehydrated creams. The majority said they consumed these products due to tradition and the taste they give food. The observed practice was to add natural seasonings by adding a small packet of bouillon powder and Lizano^®^ sauce. Participants highlighted that foods with high salt/sodium content, despite being “harmful”, were utilized due to their “ease of use”, low cost, and social consumption tendencies, as these are often advertised in television programs.


*“It would be hypocritical to say that I do not buy, when I like to make rice with chicken and put some in it, I buy canned sweet corn. Because it comes ready, so it is better. To avoid chopping, maybe and reduce effort, to avoid being in a rush.”*
(Woman, Palmar Norte)

It was discovered that in certain communities, foods with high sodium/salt content were consumed and a taste for other types of food had been acquired. The further a community is located from the country’s central region, the more some products vary, both in price and availability. In some communities of the Brunca region (the southern region of the country), the consumption of canned goods, prepared sauces, and export fruits is more common than in the central region of Costa Rica, as its closeness to the border with Panama facilitates the commerce of these products at a cheaper price than in national stores.

Through observation of the warehouses or distribution centers in the different regions—specifically in the population’s main places of purchase, such as markets, supermarkets, grocery stores, and soda shops—an ample offering of foods with high sodium/salt content at low cost was found.

The participants explained how this resulted in a rapid increase in the popularity of certain food items, such as was the case with “caldosas” (corn snacks with fish cocktail stock), which originated in Palmares (a county in the Alajuela province) and, in less than 2 years, were being sold in restaurants, soda shops, and grocery stores in the Central and Northern Huetar regions. Some inhabitants discussed how vegetarian and vegan alimentary trends had spread as ideals, implying changes in the acquisition of products and how food is prepared in these communities.

#### 3.3.2. Intent to Reduce the Consumption of Common Salt and Condiments

Female participants said they could reduce the quantity of salt added during meal preparation but could not eliminate it completely unless it was a medical recommendation—and in any case it would be difficult to abide by. Some expressions or meanings used to reference food with less than desired salt were: *“tastes like hospital food” (Woman, Liberia); “it is our means of life, we cannot live without salt because it is the flavor in food” (Woman Liberia); “it lacks love” referring to food preparation (Woman Limon); “without salt, food has no taste” (Woman, Cartago)* and *“food tastes better with salt” (Woman, Palmar Norte).* A link between flavor and the feeling of love for homemade food and the satisfaction produced when food is consumed was identified among the participating population of this study. The main sources of emotions associated with the consumption of foods with salt are shown in Table 1.

Most interviewed mothers recommended the use of traditional seasonings, such as natural seasonings, Panamanian pepper and ginger in lemon juice, “anisillo” (*Piper auritum*), and “bijagua” (*cigar plant*) leaves (in the Northern region), to provide flavor and seasoning to food, thus reducing the use of salt.

Likewise, they recognized the existence of a link between the excessive consumption of salt and certain illnesses, but did not feel at risk of contracting them, as they considered themselves to consume little salt/sodium. The study participants pointed out that excessive salt consumption can lead to high blood pressure, heart problems, and circulatory problems; among others, such as diabetes, high cholesterol, and gout (uric acid accumulation).

Some women mentioned the influence of what is cooked when certain chronic diseases like diabetes and high blood pressure are diagnosed, as it becomes necessary to make adjustments in meal preparation. The members of the family tended to rate this adjustment as being flavorless. In addition, the participants believed healthy food to have a high economic cost.

Teenagers and young adults who were interviewed reported a shift toward contemporary alimentary practices; they prepared nontraditional recipes, while others were vegan, vegetarian, or alternative in type. According to the participating population, the latter are oriented—from their point of view—to healthier lifestyles, as they associate healthy nutrition with fat-restricted diets and emphasize the importance of performing physical activity.


*“His wife prepares him a healthy snack because she loves him and does not want to be widowed early, and it turns out that at the office the guy takes out the healthy food she prepared, opens his desk drawer and pulls out the salt shaker…”*
(Woman, San Jose)

Consumers wishing to raise awareness in their households about the products they buy were among the group of young adults; they promoted reading the ingredients list of processed products. However, none indicated being interested in observing the amount of salt/sodium in food.


*“As a consumer, what do you need in order to buy products that are low on salt? I think what I am missing is to read a little more because normally almost all products have details of the composition. Well, the fact that everything is labeled makes shopping easier, to know the quantities and as I was saying, it is good to go to lectures where you are told about the good stuff and the right amounts. At least about sugars, she told us mostly about sugar not so much about salt. She told us that one spoon a day is what the body needs to fall into excess.”*
(Woman, San José)

### 3.4. Eating Outside the Home

Most participants, regardless of their age, expressed having a good income and confessed eating out more than twice a week, which was mainly driven by social engagements with friends, family, or work colleagues. On the other hand, participants who expressed having a more limited budget listed special occasion celebrations or time constraints as motives to eat out, with the immediate solution being “eating at a fast-food restaurant” or “ordering take-outs” with high sodium content, such as fried chicken, hamburgers, Chinese food, French fries with ketchup, “tacos”, and pizza, among others.

According to the participants, when they went out to celebrate, the person being celebrated selects the place and the food of their preference; they are given that treat because of the occasion. They stated that children often choose fast food establishments with playgrounds, and where they get a toy with the kids’ menu, such as a “Happy Meal^®^”. When adults are celebrated, they choose places where the food is more elaborate. Adults expressed preference for seafood restaurants and restaurants with à la carte menus. For most of the celebrations, participants reported using sauces and dressings to season meals, and consumed soda and alcoholic beverages in greater quantities than usual. Young participants said they “look for a healthy meal” so they usually visited establishments with sushi, vegetarian, or vegan food, which are mainly located in the central region of the country. They noted that in remote regions there are few establishments of this type; there are only shops with natural smoothies and mango strips (Table 2).


*“(…) and if is outside and it is dinner, it is a triple sin. I categorized it in two scenarios. One goes out to eat or one is invited to dinner at someone else’s house. Then, if one goes out to eat, maybe the healthiest choice is ordering sushi…a miso soup, a salad or a roll. Otherwise, it is probably going to be pizza, but it depends of the quantity. Depends on the quantity and personal finance. Nowadays life is hard, even if I have strong cravings, I cannot treat myself, at least not me…because you know, you need money…dammit…things are tough.”*
(Woman, Heredia)

Two main types of dining out establishments were identified: fast food sales and restaurants.

In a similar fashion, most inhabitants stated noticing an increase in the number of food establishments, especially fried chicken and “caldosas,” as they are very cheap and liked by youths in the participating communities.


*“Fried chicken, the shops are everywhere, on every corner. In Heredia, chicken is sold all over the place. Chicken is sold everywhere. Chicken is always sold with lots of condiments. Lots of condiments. Since it is sold very cheap, it is sold a lot.”*
(Woman, Heredia)

## 4. Discussion

In the present study, we reiterate that in Costa Rica—as in other countries of Latin America—women are generally still in charge of cooking and “setting the table” [32,33]. Nevertheless, the acquisition of food for the household is shared by the couple or other family members. Both activities demonstrate how cultural, social, economic, psychological, and environmental factors affect food consumption [2]. Additionally, the participants demonstrated some understanding regarding the relationship between excessive salt consumption and a person’s health status.

Food preparation at home has been linked to better diet quality, as the intake of sodium can be reduced [34,35]. Nonetheless, this study demonstrated that in Costa Rica most home prepared meals are seasoned with variable quantities of salt and condiments and a wide variety of products, some Costa Rican in origin. Despite not quantifying added salt and condiments to the foods, it was deduced that the quantity was generally high. This was because the main sources of sodium in the Costa Rican diet are common salt (60%) and condiments (13%), with potential available amounts in 2012–2013 of 2.78- and 0.61-grams sodium/person/day, equivalent to 6.95 and 1.52 g salt/person/day, respectively [18].

In Peru, most mothers use condiments when preparing food [32]; this practice has become common due to globalization [36]. The most utilized condiments differ between the two countries. While bouillon cubes and dried powder (“consomme”) are popular in Costa Rica, “ajinomoto” (a popular brand of monosodium glutamate in Peru and Costa Rica) is popular in Peru [32]. In Costa Rica, a few women mentioned the use of “ajinomoto” as a food condiment. They also used curing salts to prepare “chicharrones” (pork rinds). Curing salts are used without control; therefore, this generalized practice in some communities should be monitored due to the association of nitrites with gastric cancer [37].

The explanations provided by participants regarding home food preparation highlighted the primordial role of natural seasonings and the practice of combining them with condiments to achieve the desired flavor in a dish, similar to findings in Peru [32]. It was observed that, in the search to attain the ideal taste, people add salt and other condiments without being conscious of the quantity and type. They do not usually use a measuring spoon, nor do they keep track of how much salt and other elements with high sodium content are added to preparations. People focus on achieving flavor through a trial-and-error approach (which means they taste for flavor, adding salt, condiments, and sauces until they reach the desired flavor). When the goal is achieved, we find expressions such as “*the food is just right*”, “*this is how I wanted it*”, “and “*this is how my mom (grandma) made it*”.

In regard to new ingredients incorporated into preparations, a lack of knowledge was observed regarding the amount of salt or sodium that these contain and the need for participants to have educational competence to read and interpret nutritional labels—a tool widely used to select healthier food [38]. These new, convenient products allow women to save time and effort while giving international flavors to meals, as per their recollections. Mainly young women used them, as these products have entered the national market recently. These products should be regulated due to their high sodium content, as well as most critical nutrients associated with NCD [39].

Some people who suffered from hypertension or had relatives with this illness mentioned the importance of having a low-salt diet in order to keep blood pressure in check. In general, they recognized that excessive salt/sodium intake is detrimental to human health; however, they made inadequate use of this knowledge and were unable to explain why, reflecting inadequate biomedical knowledge. These findings are similar to those found in Peru [32] and are confirmed at a national level, as found in this study in two specific communities (Liberia and Palmar Norte) of a province in Costa Rica.

One fact to consider is that people prioritize taste over possible detrimental effects that food high in salt/sodium has on their health, as found in the present and other studies [31,40]. Most health behavior theories propose that people who have good biomedical knowledge tend to use it in order to stay healthy, a factor not apparent in the participants of the present study [41].

The present study confirmed a link between flavor and the feeling of love in food prepared at home and the satisfaction generated when consumed; this coincides with the findings of an exploratory study conducted in Argentina, Ecuador, and Costa Rica [19], as well as another done in Peru [32]. The pleasant memory of having a meal is related to the positive emotion of eating, tasting, savoring, and remembering, leading to anticipation of pleasurable emotional impact [42]. Additionally, the feeling of love as a positive emotion influenced the eating behavior of some of the participants [43,44], who perceived love in the flavor of food and its presentation. Thus, the emotional consumption of foods low in salt/sodium is done from the perspective of personal and family health care. However, when the taste of prepared foods was altered, phrases related to lack of flavor or lack of love were recorded.

During the interviews, most of the participants (72%) stated that they did not have hypertension, while 28% reported having diagnosed hypertension—a value slightly lower than the national prevalence [10]. It is possible that the remaining participants had not been diagnosed or it could be that they did not accept their condition (i.e., they were in denial). A finding of great concern was the resistance to modify eating behaviors. This change is usually reactive, rather than proactive, as the population only considers taking measures when a medical diagnosis of hypertension is issued. Therefore, it can be argued that public health strategies specifically focused on salt reduction and broader strategies for healthy eating may be successful in increasing consumer awareness and knowledge about a particular issue, but remain limited in achieving a real change in behavior [45].

When participants of the community spoke about changes to their diet and cooking practices, this was often due to changes associated with age, lifestyle, or generation. There was concern among the participants about the change in flavor caused by a reduction of the salt/sodium content, which requires time for the consumer’s palate to adjust. These synergistic efforts between the communities and the Costa Rican food industry could enhance the success of future social marketing interventions [46].

Most of the interviewed participants acknowledged the frequent use of condiments to enhance the flavor of their meals; thus, learning about the use of condiments in the culinary culture should be considered a strategic measure for reducing salt/sodium consumption [47]. In the context of the USA, formative research of the “Skip the Salt, Help the Heart” campaign showed that people associated a low salt/sodium diet with flavorless food [48]. The authors pointed out the importance of using messages to highlight an immediate exchange of benefits, such as saying the flavor will be great, instead of focusing on the importance of regularly consuming less salt to prevent the development of hypertension. This experience is essential to the efforts that must be made in Costa Rica and shows that, despite cultural differences, some factors hold true when the target population has a great affection for salt/sodium.

As salt is not the only source of sodium, social marketing strategies could also highlight the importance of learning to moderate the use of sodium products, such as curing salts and monosodium glutamate. Salt measurement would allow people to be aware of when they are using too much salt and to consider reducing it.

Therefore, those responsible for public health policies must change their approach and methodology. This could consist of carrying out several strategies simultaneously, such as social marketing, emphasizing the physical visualization of the recommended 5–6 g of table salt per day and the dangers of food types with hidden excess salt/sodium, as well as providing education on how to interpret the sodium/salt content from the labels of processed products. Furthermore, it is important to discourage the excessive addition of salt/sodium, condiments, breading, dressings, bouillon (chicken, beef, and shrimp), bouillon cubes, instant soups, dehydrated creams, commercial sauces (vegetable or Lizano^®^ sauce, mayonnaise, and tomato sauce), seasonings, and flavored salts to meals. Finally, education about the health risk of the use of curing salts, which have been associated with certain types of cancer, should be facilitated [49,50].

People should also have available alternatives to salt and learn how to make their food tasty and perceived as gourmet using locally available herbs such as garlic, lemon, sour orange, onion, oregano, basil, and cilantro. This becomes another challenge, due to the wide and varied availability and accessibility of existing salts in the communities participating in the present study (e.g., garlic, onion, herbs, lemon, and smoked flavored salts). These flavored salts may have influenced the evolution of Costa Rica’s culinary culture, which has become globalized [51], as evidenced in studies carried out in other latitudes [52,53].

Another important issue to take into consideration is the ignorance of the population about salt content in food that does not have a salty flavor. In developed countries, researchers have found that many people did not recognize hidden salt in processed food [54]. By contrast, the participants in this study knew that salt was added to various processed products and food types, but the majority believed that sweet foods and products did not contain salt/sodium, and were accustomed to balancing sweet and acidic flavors with salt to improve the flavor.

Social marketing and communication in Costa Rica should be directed toward promoting the reduction of salt/sodium consumption in Costa Rican households, taking into account the following items: negative attitudes towards diet changes, the value of taste/flavor in food, the notion of preparing food with “love”, the practicality of achieving desired flavors, the low availability of certain ingredients in some areas of Costa Rica, and economic limitations.

Therefore, the strategy should be focused on the positive aspects of reducing salt/sodium consumption, such as health improvement with an emphasis on preventing high blood pressure. Even though the participants made this association, they did it in a weak fashion at best; not strongly enough to drive the desired behavioral change. Some participants even perceived hypertension as a disease that is common among people and not very serious when compared with diabetes or cancer. They believed that high blood pressure can be controlled by just taking a pill, while diabetes is painful and complicated, as one needs to take insulin shots and may eventually suffer from limb amputation.

### Strengths and Limitations

The strengths of the current research include its national scope and accountability for cultural diversity concerning food preparation practices and perceptions about the use of salt, seasonings, and condiments. The socioeconomic status of the participating population predominates in Costa Rica; therefore, the results on food preparation and acquisition reflected what happens in a majority of Costa Rican families during the period of study.

A significant limitation of the present study is that most of the participants were women, given their availability at home during the daytime when the research team made their visits; therefore, it is necessary to explore the perspectives of men in more depth. Some studies have shown that men use health services less than women and have less knowledge of their health status [55,56]; however, this could not be confirmed in the present study. In addition, some sociodemographic data and family characteristics were based on self-reporting, which may be influenced by social convenience and memory bias.

## 5. Conclusions

The addition of excess salt, as well as other industrialized ingredients with high sodium content, was found to be characteristic of the preparation of food, both at home and outside the home, in Costa Rica.

Beyond the cultural and geographical differences in Costa Rica, age factors were suggested as being the main differentiators, in the use of salt, seasonings, and condiments for home food preparation, the type of recipes prepared, and in the selection of the type of establishment to eat out at (the latter was affected by economic aspects as well).

Most participants did not have a clear understanding of their daily salt/sodium intake, as they could not calculate the salt added during food preparation or hidden in a variety of foods, including sweet flavored and processed products.

Finally, the present study revealed that salt is an essential component of foods with strong cultural roots. Its flavoring function in food preparations is associated with emotions and caring of the family. The deeply rooted values and meanings associated with its use in food indicate that the implementation of salt reduction strategies in Costa Rican communities is challenging, requiring scientific precision in designing community interventions.

## Figures and Tables

**Table 1 ijerph-18-01208-t001:** The five sources of emotions associated with the consumption of homemade food with salt/sodium.

Source of Emotions Associated with Food	Feeling Identified in the Study
Sensorial properties	Tastes like mom’s food Has a pleasant look and taste I use Lizano^®^ sauce and margarine to give taste Salt is the flavor
Experiential consequences	Eating gives me a sensation of pleasure
Associated consequences	You eat with little salt to prevent high blood pressure You try to eat as naturally as possible to have good health
Personal consequences or personal significance	To prepare food this way reminds me of family Using a condiment or processed seasoning reminds me of my mother/grandmother
Behavior of involved agents	If food has good taste, they feel happy If food lacks salt (taste), the family resents that I cook with little salt due to medical recommendations

**Table 2 ijerph-18-01208-t002:** Decisive factors in the selection of the dining establishment by age group reported by the key informants.

Factor	Age Group		
	Over 35 Years	Young Adults	Teenagers and Children
Physical accessibility	Widely available low-cost fried chicken options. Constitutes a viable option for families for any meal.	Diversity of restaurants located close to the workplace. Going out to eat in groups is an everyday social practice. Trying something new is a temptation.	Frequent fast-food establishments close to school, with refill service and low-cost menus (combos with hamburger, French fries with ketchup, and soft drinks).
Tastes and preferences	Prefer typical food or seafood restaurants. Like establishments located in the local markets. Buy commercial Chinese food often (chow mein and chop suey with soy sauce), mainly on the weekends.	Like trying different food, mainly on the weekends and celebrations. Enjoy the experience of eating in Japanese, vegetarian, vegan, and international restaurants, as they are considered innovative.	Seek fast food establishments (food courts in malls, smoothie, ice cream, or beverage shops), even if it implies traveling longer distances. Buy fruits seasoned with salt, lemon, vinegar, and Lizano^®^ sauce from street vendors. Children prefer fast food for the food, onsite playgrounds, and kids’ meal toys.

## Data Availability

The data presented in this study are available on request from the corresponding author after authorization by the ethical committee of INCIENSA. The data are not publicly available due to ethical reasons.

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
