# Peer review of "Household Cooking and Eating out: Food Practices and Perceptions of Salt/Sodium Consumption in Costa Rica"

_ijerph, 2021, doi:10.3390/ijerph18031208_

Round 1
Reviewer 1 Report
The paper lacks cultural and geographical differences in Costa Rica, which are mentioned even in conclusions. The study was carried out in six regions. It would be interesting to know about food and cooking differences in those regions.
83- “Community members were selected considering their knowledge on the food supply in the region”. What does it mean? They were interviewed before the survey?
87 –“9 cultural managers and 59 key informants”. In 136,137,138 it is written, that “Cultural managers, through their work, know the socio-cultural characteristics of the population as they carry out activities focused on cultural management, artistic-cultural expression, entrepreneurship and revitalization of tangible and intangible cultural heritage”. According to this, their knowledge about cooking traditions and eating habits in Costa Rica have to be better. But all the results are presented as one group answers.
130-150. It would be better to present average age of participants with min and max age. From this part we can realize, that students (13 %) also were interviewed. In 354 the teenagers are mentioned. But there are no results about this particular group. It would be interesting to know the age of this group.
396 – in the table there is a column „Teenagers and children“. What does it mean? That children also participated? Or in this table we can see the participants opinion about eating habits of different groups?
Reviewer 2 Report
Submitted manuscript "Household cooking and eating out: consumption of salt/sodium in Costa Rica" aims to study food preferences and consumption of salt in household cooking and “eating out of home” practice. The interview/workshop material used for this manuscript is very interesting. Discussing specific geographical food preferences and personal feelings about cooking in Costa Rica is curious and will be of interest for journal’s readers. However, the way this manuscript is written is not optimal from the scientific point of view and must be strongly improved.
Keywords. Authors use keywords that are not relevant to presented in the manuscript results: cardiovascular disease, high blood pressure, hypertension... all these parameters we not measured or directly accessed by the co-authors and are speculative.
Introduction contains general expressions about importance of salt reduction and does not reflect Results and Conclusion sections. Several important References referring to the studies about salt consumption already performed in Costa Rica are missing in the Introduction section.
Results section is not connected to Introduction and research questions blur out. Title of the manuscript "Household cooking and eating out: consumption of salt/sodium in Costa Rica" does not reflect Results. For example Section 3.3. "Eating out of home" does not even mention salt/sodium, but discusses decisive aspects why participants chose to eat out of home.
In general, this manuscript reassembles efforts to identify eating patterns within certain parts of Costa Rica population (geographical area and… women?) where salt is somewhat taken out of the pattern context. Moreover, there was no possibility to quantify salt/sodium used for cooking, therefore conclusions about “high salt” of “salt reduction” are somewhat speculative. Moreover, scientific literature until know does not support a complete abundance of salt and it is also not surprising that interviewed persons mentioned “without salt, food has no taste” or “food tastes better with salt”.
Discussion section is well written and is easy to follow. Arising question is whereas interview participants or their family members have existing problems with blood pressure/ heart diseases which could make a conclusion about high salt content in house cooking more clear and also discussion about knowledge “high salt is unhealthy” more sinful.
Reviewer 3 Report
This manuscript addresses an important topic: sources/aspects of salt intake in food preparation and purchase in Costa Rica. The research, however, is not well contextualized with other related reports; the innovation(s) are not specified; the methods are not provided in detail; the English is disfluent; etc.
This manuscript has distracting English disfluencies and needs to be edited by a professional language Editor.
Abstract:
Line 16: please state the four qualitative research techniques.
Line 16: Data is plural.
Line 17: This is usually referred to as the Social Ecological Model.
The Discussion reveals there is a substantial literature on sources of sodium/salt in the diet and cooking and eating out practices. This needs to be briefly reviewed in the Introduction, which needs to end with a brief statement about what is innovative about the reported research.
2.1 Anthropology-ethnology covers many types of studies. Please provide a more detailed statement/description of research design. For example, the Abstract says four qualitative methods were employed. Please state and briefly describe them with references.
Location selection criterion b says communities were selected which had the highest prevalence of stroke mortality. Shouldn’t the selected communities have included some with low stroke mortality so that comparisons can be made to identify differences between those that do and do not likely contribute to stroke? For example, if practices were high/common in high and low communities, they probably aren’t related to stroke. Please address.
What are cultural managers? That sounds totalitarian? What do they do? What training do they have?
A table is needed of the demographic characteristics of the participants, probably for cultural managers and key informants separately.
Line 23: What the authors refer to as the Ecological Model is usually labeled as the Social Ecological Model. Ref 26 specifies the Transtheoretical Model, not the Social Ecological Model. This needs to be revised.
Lines 90-4: Please move to 2.1 and provide an example of how each method was employed.
Lines 119-20: There are many different ways of analyzing qualitative verbal transcripts. Non-Spanish speaking readers will not be familiar with speech or content analyses methods (ref 26-7). Thematic analysis is the most common method employed/reported in the English language, but alternatives exist. Please specify the closest method of qualitative verbal transcript analysis with English terms and references.
Lines 87-8 state there were 68 participating adults, but lines 131-2 report 11 cultural experts and 80 community members. Which was it? Please be precise and consistent.
Line 138: Why would experts in artistic-cultural expression and entrepreneurship know anything about food purchase and preparation practices or dietary intake?
It is hard to imagine that a substantial number of adults ages 35-65 years would not have non-communicable diseases (lines 149-50). Please address.
If 72% of participants did not have high blood pressure, is it possible they did not practice the salt intake related behaviors at high enough levels to lead to hypertension, and thereby would provide misleading data? Please address.
Line 154: Who were the other 50%?
The Discussion rambles. It would be helpful for the first paragraph in the Discussion to specify the 5 or 6 or so key findings from your research. Each ensuing paragraph should provide more detail to support that finding and state how national policy might address it.
Round 2
Reviewer 2 Report
The manuscript was revised and new version is significantly improved. I reccomend to avcept the revised manuscript.
Author Response
Point 1. Reviewer 2 comment was: The manuscript was revised and new version is significantly improved. I recomend to accept.
Response 1. Thanks
Reviewer 3 Report
The English, at times is hard to follow, still.
Each of the four research/data collection methods are not described in detail.
Why was a study conducted in pilot mode in 2011, and conducted in 2016, just submitted in 2020? Would social changes since then have transformed the information thought to be obtained?
Line 76: KAP usually refers to Knowledge, Attitudes and Practices. It is more usually referred to as KAB: Knowledge, Attitudes and Behaviors.
I am not sure what the authors believe was learned from their data? It would be helpful for the authors to summarize what they learned in a paragraph at the end of each numbered section in 3.3 and 3.4.
The Results report usual dietary practices. The Discussion spends a lot of space on intervention. There is a huge conceptual leap from description to inferences about intervention. There was very little about intervention in the Methods or Results. The authors should change their suggestions about intervention to research questions that need to be tested?
I didn't understand lines 575-7.
Lines 594-8 is a run-on sentence and needs to be broken into subparts to be understood.
Author Response
Comments and Suggestions for Authors
Response to Reviewer 3 (Round 2)
Point 1. The English, at times is hard to follow, still.
Response 1. In addition to specialized edition done by MDPI's English editing service, an English-speaking colleague that lives in Costa Rica and knows well its food culture (MS, CNW Marie Elena Hawkins), edited the manuscript. You can verify its improvement.
Point 2. Each of the four research/data collection methods are not described in detail.
Response 2. The data collection techniques have its procedure indicated in the references. However, they were describied as applied in the manuscript as follows:
1) Semi-structured interview: the interviewee had the opportunity to express their points of view in an open remark interview situation. 2) Investigative workshop: offered the ability to explore the research topic from comprehensive and participatory perspectives through discussion, reflection, and group feedback. The purpose of these workshops was to complement and deepen the information obtained from the interviews with cultural experts and key informants from the communities. The workshops were conducted by an anthropologist, further facilitating the study investigation. Each workshop consisted of an introductory section, such as the explanation of the dynamics and individual presentation, or the recapitulation of the previous workshop; a development section and a closing activity. 3) Demonstrations: were carried out by the participants who agreed to prepare dishes that they usually eat at home as a family. This allowed the researchers to verify what was reported during the interviews and workshops, and to identify other details surrounding culinary practices and family eating habits. 4) Participant observation: allowed the researchers to methodically systematize the information they observed, overheard, and otherwise gathered regarding daily practices and processes of salt/sodium consumption. Research team observations were made and recorded according to guides developed specifically for each technique, as well as the characteristics of the participants.
First, interviews were carried out and systematized with cultural experts. Workshops and interviews with key informants were then developed, and observations were made during both. Last, demonstrations were conducted, with those participants who agreed to perform them. The four techniques allowed a progression from the description of the data, to reflection and action, on through the prioritization of facts and needs, culminating in the formation of decisions supported by the majority of the participants.
Point 3. Why was a study conducted in pilot mode in 2011, and conducted in 2016, just submitted in 2020? Would social changes since then have transformed the information thought to be obtained?
Response 3. Both studies were carried out when external funds (grants) were available. Costa Rica is an LMIC, that is, with limited resources to carry out an investigation of this type. The delay in the publication of the results was due to the fact that the research team obtained a new grant for a consortium of countries project. It was a wide, complex and demanding project that just finished. In addition, it included among its objectives the deepening of certain aspects of eating practices and perceptions to generate a social marketing plan to reduce salt intake in a segment of the population. Therefore, the team did not have the time available to write a manuscript and follow the publication process. During this formative research, whose results will be published, similar results were found in the aspects that both studies agreed.
It is not considered that there were relevant social changes in this matter that could transform the information obtained during this period. The leading team of the reduction of sodium consumption program at the national level is ours and therefore other groups haven’t done interventions. Furthermore, substantive social change generally does not occur in such a short time.
Point 4. Line 76: KAP usually refers to Knowledge, Attitudes and Practices. It is more usually referred to as KAB: Knowledge, Attitudes and Behaviors.
Response 4. KAP was replaced by KAB.
Point 5. I am not sure what the authors believe was learned from their data? It would be helpful for the authors to summarize what they learned in a paragraph at the end of each numbered section in 3.3 and 3.4.
Response 5. The researcher team learned in this research the following: regardless of the place of residence, socioeconomic group, cultural diversity, educational level, and sex; intergenerational differences are the main determinants of the foods that are acquired, prepared, and use of culinary salts and other flavorings (condiments, sauces and dressings, among others), of how they are seasoned at home, as well as of the foods that are consumed outside of it and the dinning places. This suggests that the offer and advertising of food, and the lifestyles of the participants, modulate these practices and perceptions in different age groups. We use the results of the investigation to select the target segment and design a formative investigation that has already been carried out (see papers objective). All this information is discussed and forms part of the conclusions reached by the authors. Therefore, it is not considered necessary to add to the paper this information because it would be repetitive and the article is already extensive.
In summary, in section 3.3, which corresponds to cooking at home, the findings allowed us to understand in more detail and depth, the practices and perceptions of the participants according to age group, sex and region of residence, regarding various aspects of food at home and outside, particularly associated with salt/sodium. For example, who and what food they buy, and why; who are responsible for preparing them; when, how and with what they are seasoned and give the final point of flavor to the preparations; what is the culinary use of common salt and other salts; that they perceive the excessive consumption of salt and the intention to reduce it; which foods they consider as sources of salt / sodium; what are the eating patterns associated with the consumption of salt / sodium; which foods high in sodium are consumed more frequently; for special celebrations what foods do they prepare and how; the association between emotions and consumption of homemade food with salt/sodium; which factors in the social and cultural environment influence healthy decision-making related to salt / sodium consumption. Likewise, it allowed to verify and find unexpected results.
In section 3.4, which corresponds to the consumption of food outside the home. Decisive factors in the selection of the dining establishment by age group were identified. In addition, it was verified that the foods selected by some age groups are due to taste and accessibility. These are foods high in salt. It is confirmed that the age group is the leading factor in the selection of food, both eating outside or at home.
Point 6. The Results report usual dietary practices. The Discussion spends a lot of space on intervention. There is a huge conceptual leap from description to inferences about intervention. There was very little about intervention in the Methods or Results. The authors should change their suggestions about intervention to research questions that need to be tested?
Response 6. The paper is not about an intervention, it is scientific research. Its objective was to study the dietary practices and perceptions related to the excessive consumption of salt and sodium, at home and outside the home, in communities of all regions of Costa Rica, including various cultural practices of the population's diet. The findings derived from the responses of the participants served as input to identify the sociodemographic profile of the target segment and design formative research questions to build a communication and social marketing plan to reduce the excessive consumption of salt and sodium. Therefore, the research questions previously proposed were answered in this paper. Furthermore, the methods and results sections do not include aspects related to an intervention, as it was not the purpose of the study.
Point 7. I didn't understand lines 575-7.
Response 7. Ok, it was corrected.
Point 8. Lines 594-8 is a run-on sentence and needs to be broken into subparts to be understood.
Response 8. Ok, it was corrected.
|
20 October 2020 |
Submission Date |
|
08 Jan 2021
|
Date of this revie |